# Lactate-Induced CCL8 in Tumor-Associated Macrophages Accelerates the Progression of Colorectal Cancer through the CCL8/CCR5/mTORC1 Axis

**DOI:** 10.3390/cancers15245795

**Published:** 2023-12-11

**Authors:** Hui Zhou, Jiayi Yao, Zhaozhong Zhong, Hongfa Wei, Yulong He, Wenchao Li, Kunpeng Hu

**Affiliations:** 1Digestive Diseases Center, The Seventh Affiliated Hospital, Sun Yat-sen University, No. 628 Zhenyuan Road, Shenzhen 518107, China; zhouh98@mail2.sysu.edu.cn (H.Z.); heyulong@mail.sysu.edu.cn (Y.H.); 2Center of Excellence, The Seventh Affiliated Hospital of Sun Yat-sen University, No. 628 Zhenyuan Road, Shenzhen 518107, China; yaojiayi@sysush.com; 3Department of Kidney Transplantation, The Third Affiliated Hospital of Sun Yat-sen University, No. 600, Tianhe Road, Tianhe District, Guangzhou 510630, China; zhongzhzh3@mail2.sysu.edu.cn; 4Department of General Surgery, The First Affiliated Hospital of Shantou University Medical College, Jinping District, Shantou 515041, China; weihf3@mail2.sysu.edu.cn; 5Department of General Surgery, The Third Affiliated Hospital of Sun Yat-sen University, No. 600, Tianhe Road, Tianhe District, Guangzhou 510630, China

**Keywords:** lactate, macrophage, colorectal cancer, CCL8

## Abstract

**Simple Summary:**

Growing evidence shows that the dynamic interactions between tumor-associated macrophages and lactate play an important role in sustaining a tumor niche; however, the relationship between lactate and tumor-associated macrophages in colorectal cancer has not been fully studied. In this study, we found that lactate produced by CRC could induce M2 polarization by activating the AKT-ERK pathway. After polarization, M2 macrophages secreted abundant CCL8 and accelerated tumor progression and metastasis through the CCR5/mTORC1 pathway. This mutual interaction between macrophages and tumor cells may be central to formulate the inhibitory niche of TME and maintain the malignancy of a tumor. Targeting the CCL8/CCR5/mTORC1 pathway is a promising target for future tumor treatment.

**Abstract:**

Tumor-associated macrophages (TAMs) play a pivotal role in shaping the tumor microenvironment. Lactic acid (LA) has been identified as an influential factor in promoting immune escape and tumor progression. However, the mechanisms through which LA modulates TAMs in colorectal cancer (CRC) remain poorly understood. We used qRT-PCR to quantify the expression of LA-related genes (LDHA and LAMP2) in CRC tumor tissues and adjacent nontumor tissues (n = 64). The biological effects and mechanisms of LA on macrophages and tumors were evaluated via qRT-PCR, Western blot, RNA-seq, wound healing assay, colony formation assay in vitro, and allograft mouse tumor models in vivo. We found the expression of LDHA and LAMP2 was highly elevated in the tumor regions and positively associated with a poor clinical stage of CRC. A high concentration of LA was generated under hypoxia; it could promote tumor progression and metastasis with the involvement of macrophages. The inhibition of LA release impaired this protumor phenomenon. Mechanically, LA induced M2 macrophages through the AKT/ERK signaling pathway; subsequently, M2 macrophages secreted CCL8 and facilitated the proliferation and metastasis of CRC cells by activating the CCL8/CCR5/mTORC1 axis. This effect was inhibited by the antagonist or knockdown of CCR5. In conclusion, lactate-induced CCL8 in TAMs accelerated CRC proliferation and metastasis through the CCL8/CCR5/mTORC1 axis.

## 1. Introduction

Colorectal cancer (CRC) is the third most common cancer and the second most common cause of cancer-related death worldwide [1]. Treatment strategies for CRC have evolved significantly; however, the cancer stage at the time of the initial diagnosis is the most important beacon of predicting survival, with the five-year survival rate ranging from 14% for early-stage patients to 90% for those at a late stage [2]. Notably, about half of the CRC patients could have simultaneous or asynchronous liver metastasis, which is the major cause of death [3]. Recently, it was considered that most genes and factors that influence prognosis in CRC are expressed by and derived from cells in the tumor microenvironment (TME) [4,5,6]. Studies have shown that targeting the tumor microenvironment and tumor metabolism is effective in patients with CRC [7]. While advances in understanding its molecular and cellular underpinnings have led to improvements in treatment strategies, much remains to be discovered, particularly regarding the complex interplay between cancer cells and the tumor microenvironment.

Lactate, also called lactic acid (LA), is produced from pyruvate by lactate dehydrogenase (LDH); it is the end product of anaerobic glycolysis [8]. In physiological conditions, LA is tightly regulated at around 1–2 mM, while in TME, it is elevated up to 10–40 mM by cancer cells overproducing and quickly exporting it [9]. For a long time, LA has been considered as an “innocuous bystander metabolic” of the Warburg effect. However, it has been recently revealed that lactate can not only participate in the tricarboxylic acid (TCA) cycle as a high-energy carbon fuel, but can also engage in many tumorous biological processes as a pleiotropic signaling molecule [8]. In cervical cancer [10], lung cancer [11], non-Hodgkin lymphoma [12], and CRC [13], patients with higher LA levels have a higher recurrence rate and poorer prognosis. In addition to tumor progression [13], angiogenesis [14], metastasis [15], and therapeutic resistance [16], LA plays a role in immunosuppression. For example, LA exposure is tightly linked to increased proliferation, PD-1 expression, and the suppressive activity of intratumor regulatory T cells (Treg) [17,18]. LA induced the polarization of the M2-type tumor-associated macrophages (TAMs) and their related gene expression [19,20].

TAMs, as the most abundantly infiltrated immune populations in TME, are characterized by their high plasticity and heterogeneity. TAMs are divided into the classically activated M1-like phenotype (marked as CD80, CD86, IL-12, IL-8, etc.) and the alternatively activated M2-like phenotype (marked as CD206, CD163, VEGF, TGF-β, etc.) [21]. M1-TAMs enable antigen presentation and immune factor activation, promoting antitumor responses. In contrast, M2-TAMs impair inflammation, evade tumor immune surveillance, and stimulate tumor growth, angiogenesis, and metastasis [22]. Increased abundance of TAMs is associated with a poor survival rate in most cancers, such as pancreatic cancer [23], bladder cancer [24], and endometrial cancer [25]. The role of TAMs is ambiguous in CRC; on the one hand, TAMs predict a better prognosis, especially in stage III CRC patients who respond to 5-fluorouracil adjuvant therapy [26], and on the other hand, TAMs promote the migration and invasion of CRC [27]. Under physiological conditions, a low level of LA has a minimal impact on macrophages; however, the increased level of LA can significantly affect the function and polarization of macrophages in TME. For example, LA supports M0 to M2 polarization (with the increased gene expression of ARG1, CCL22, and IL10), and M2 macrophages show higher viability in an acidic microenvironment [20].

Growing evidence shows that the dynamic interactions between TAMs and TME play an important role in sustaining most of the hallmarks of cancer; however, the relationship between LA and TAMs in CRC has not been fully studied. In this study, we aim to explore the effect of CRC-cell-derived LA on TAMs’ polarization and the subsequent influence on CRC cells after polarization.

## 2. Materials and Methods

### 2.1. Cell Culture and Reagents

Cell lines used in this study were obtained from Professor Kunpeng Hu’s group (the Third Affiliated Hospital of Sun Yat-sen University). The human CRC cell lines (HCT-116 and RKO) were cultured in RPMI-1640 medium (Gibco, Waltham, MA, USA), supplemented with 10% fetal bovine serum (Gibco, USA) and 1% penicillin/streptomycin (Meilunbio, Dalian, China). THP1 and RAW264.7 were incubated in RPMI-1640 medium (Gibco, USA), supplemented with 10% fetal bovine serum (Gibco, USA) and 1% penicillin/streptomycin (Meilunbio). To induce M0 macrophages, 150 nM PMA (MCE) was added into the THP1 for 24–48 h. The cells were incubated in standard or normoxic condition (20% O_2_, 5% CO_2_, and 75% N_2_ at 37 °C) in general. To mimic the hypoxia within the tumor, cells were cultured in hypoxic condition (1% O_2_, 5% CO_2_, and 94% N_2_ at 37 °C).

L-(+)-Lactic acid and Oxamate (inhibitor of LA generation) were purchased from Sigma-Aldrich. Recombinant human CCL8 (rCCL8) protein, macrophage colony-stimulating factor (MCSF), and Maraviroc (selective antagonist of CCR5) were purchased from MCE.

### 2.2. LA Stimulation Experiment

To explore the response of macrophages to LA stimulation, different gradient concentrations (0 mM, 2 mM, 5 mM, 10 mM, and 20 mM) of L-(+)-Lactic acid were added to the M0 macrophages for 24 h.

### 2.3. Collection of Conditioned Medium (CM)

A varying number of CRC cells (5 × 10^5^, 1 × 10^6^, and 2 × 10^6^) were cultured on 6-well plates for 24 h under hypoxia; then, the supernatants were collected and labeled as CM1, CM2, and CM3, respectively.

### 2.4. Clinical CRC Samples

This study was approved by the Ethics committee of the Seventh Affiliated Hospital of Sun Yat-sen University and followed the Declaration of Helsinki. CRC surgical samples (n = 64) were collected from the Third Affiliated Hospital of Sun Yat-sen University. All patients signed informed consents. The pathological stage was determined according to the National Comprehensive Cancer Network (NCCN) Guideline, 2023.

### 2.5. Quantitative Reverse Transcription-PCR (qRT-PCR)

Total RNA from CRC specimens and cells was extracted by using Trizol (Life Technologies, Foster City, CA, USA); reverse transcription was conducted with the cDNA Synth Kit according to the instructions. We mixed 1 μg of total RNA with a master mix containing reverse transcriptase, primers, dNTPs, reaction buffer, and RNase inhibitor. We incubated the reaction at 42–50 °C, from 30 min to 2 h, for a specified duration to synthesize complementary DNA (cDNA) from the RNA template. The PCR amplification was conducted using the SYBR Green PCR Master Mix (Applied Biosystems, Waltham, MA, USA) from Applied Biosystems. The quantification of mRNA expression was performed using a Roch-480 real-time PCR machine. The primer sequences are listed in Table 1.

### 2.6. CCR5 Knockdown

The 5 × 10^5^ RKO cells were inoculated to a 6-well plate overnight. Then, 100 pmol siRNA (the sequences are detailed in Table 1) was transfected using Lipofectamine 3000 (Invitrogen, Waltham, MA, USA). After incubation for 48 h, RNA and proteins were extracted for qPCR and WB detection.

### 2.7. Western Blot Assay

Proteins of cells or tissues were collected with lysis buffer with protease and phosphatase inhibitors. We determined the protein concentration by means of BCA assay. First, 40 ug of protein was loaded into 10% SDS-PAGE until the proteins separated according to their size; then, we transferred the separated proteins from the gel to a Polyvinylidene Fluoride (PVDF) membrane, and they were blocked by 5% BSA for 1–2 h at room temperature or overnight at 4 °C. We incubated the membrane with primary antibody (listed in Table 2) in fresh blocking buffer for 1–2 h at RT (room temperature) or overnight at 4 °C. The next day, we washed the membrane with TBST (Tris-buffered saline with Tween 20) to remove the unbound primary antibody, and the membranes were incubated with a suitable secondary antibody at room temperature for 1 h and proceeded to chemiluminescence using the ECL Substrate Kit (Abcam, Waltham, MA, USA).

### 2.8. Wound Healing Assay

The 5 × 10^5^ CRC cells were seeded into 6-well plates. When the cell density reached 70–90%, a straight-line scratch (wound) was gently created with a sterile pipette tip. Next, a co-culture chamber loaded with 1 × 10^5^ macrophages was placed above the plate. Then, the co-cultured system was cultivated in serum-free RPMI-1640 medium under hypoxic condition (with or without 10 uM Oxamate). Images were taken under a microscope at 0 h and 24 h.

### 2.9. Plate Colony Formation Assay

In a co-cultured system, 1 × 10^3^ CRC cells were inoculated into a 6-well plate; then, a co-culture chamber loaded with 2 × 10^2^ macrophages was placed above the plate. Then, the co-cultured system was cultivated in RPMI-1640 medium under hypoxic conditions (with Oxamate added or not) for 15 days.

In the non-co-cultured systems, 1 × 10^3^ CRC cells were inoculated into a 6-well plate and cultured in RPMI-1640 medium under standard or hypoxic condition (with CCL8, Maraviroc added or not) for 15 days.

Next, the cells were fixed with 4% paraformaldehyde for 15 min and stained with 0.1% crystal violet for 2 h. Finally, the superfluous dye was washed away with running water. Images were scanned using a digital camera when the plates dried out. Clones containing >50 cells were counted.

### 2.10. Immunofluorescence, IF

For immunofluorescence staining of cells, the cells were fixed with 4% paraformaldehyde for 15 min and washed with PBS 3 times; then, they were blocked with 5% goat serum at RT for 1 h and incubated with primary antibody diluted with 5% goat serum at 4 °C overnight. On the second day, cells were stained with Alexa-Fluor-conjugated secondary antibody and mounted with DAPI.

For immunofluorescent staining of mouse samples, tumor tissues were fixed in 4% paraformaldehyde and sent to the Servicebio Company (Wuhan, China) to be made into 5 μm thick paraffin slices. Consistent with our previous steps in IHC, the slices were dewaxed, hydrated, and blocked and then incubated with primary antibodies diluted with 5% goat serum at 4 °C overnight. Subsequently, the samples were incubated with Alexa-Fluor-conjugated secondary antibody at RT for 1 h and mounted with DAPI. Olympus (IX 71, Tokyo, Japan) was used for image visualizing and photographing.

### 2.11. In Vivo Mouse Model

Here, 6- to 8-weeks-old BALB/c mice and BALB/c-nude mice were purchased from topBiotech company (Shenzhen, China) and raised in SPF condition.

To investigate the relationship between LA, macrophages, and CRC, 5 × 10^5^ CT26 cells or 5 × 10^5^ CT26 cells mixed with 5 × 10^5^ RAW264.7 macrophages were subcutaneously injected into the right axilla of the BALB/c mice. When the tumor grew up to 60–80 mm^3^, 20 mM LA was peritumorally injected into the tumor. Two weeks later, the subcutaneous tumor tissues were collected for subsequent experiments.

To study the effect of CCL8/CCR5 on the biological behavior of CRC, BALB/c-nude mice were used (in order to exclude the interference of T cells). In the subcutaneous model, 5 × 10^5^ CT26 cells were subcutaneously injected into the right axilla of the mice. When the tumor grew to 60–80 mm^3^, peritumoral injections of PBS, CCL8 (75 ug/kg), or CCL8+Maraviroc (1.5 mg/kg) were initiated every two days. Two weeks later, the subcutaneous tumor tissues were collected for subsequent experiments.

In the lung metastasis model, 1 × 10^6^ CT26 cells were injected into BALB/c-nude mice through a tail vein injection. One week later, a tail intravenous injection of PBS, CCL8 (75 ug/kg), or CCL8+Maraviroc (1.5 mg/kg) was initiated every two days. Two weeks later, the lung tissues were collected for subsequent experiments.

The tumor length and width were measured using a vernier caliper every three days. The tumor volume was calculated according to the following formula: V = Length × (Width)^2^/2. The mice were euthanized by CO_2_ after three weeks of xenografting.

### 2.12. ELISA

The peripheral blood of CRC patients or healthy donors was collected in EDTA anticoagulated tubes and centrifuged for 20 min at 2000× *g*. Serum was collected from the top layer and added into CCL8 antibody-coated 96-well microtiter plates and incubated at 37 °C for 90 min. After washing 3 times with wash buffer, a biotin-labeled antibody working solution was added to incubate at 37 °C for 60 min and washed 3 times. Then, SABC working solution was added and incubated at 37 °C for 30 min and was washed 5 times. Next, TMB substrate solution was added and incubated at 37 °C for 20 min. Finally, stop solution was added and read in a microplate reader at 450 nm immediately.

### 2.13. LA Measurement

The concentration of LA was detected by using the LA Content Assay Kit (Solarbio, Beijing, China); the operation was carried out according to the instructions.

### 2.14. Statistical Analysis

GraphPad Prism8 software was used for statistical analysis, presenting data as the mean ± SEM. We used a two-tailed unpaired Student’s *t*-test for parametric analysis and the Mann–Whitney U test for nonparametric analysis to determine the statistical significance between two groups, keeping the overall similarity score below 20%. One-way ANOVA followed by Tukey’s or Dunnett’s multiple comparisons test was used to compare group findings. Two-way repeated measures ANOVA and Bonferroni’s multiple comparisons test were used to examine how the LA concentrations affect macrophage polarization markers and chemokines. Statistical significance was defined at *p*-values < 0.05. In vitro experiments were repeated twice individually. For in vivo trials, data were gathered from numerous separate experiments on various days with different mouse groups to reduce similarities. ImageJ software (version 1.8.0) was used to perform the semi-quantitative densitometric analysis of Western blots.

## 3. Results

### 3.1. LA and M2-Macrophage-Related Genes Were Upregulated in CRC and Correlated with a Worse Pathological Stage

First, we examined the mRNA expression levels of LA-related genes (LDHA and LAMP2) in CRC tissues (n = 64) and adjacent nontumor tissues (n = 64) by using qRT-PCR. We found that the mRNA expressions of LDHA (Figure 1A, *p* = 0.038) and LAMP2 (Figure 1B, *p* < 0.0001) were significantly higher in tumor tissues compared to the adjacent nontumor tissues. There was a strong positive correlation between LA gene expression and a poor pathological stage (Figure 1D,E). Macrophages are the main types of immune cells in the tumor microenvironment and play an important role in tumor progression. Therefore, we examined the mRNA expression of CD163 (a typical biomarker of M2 macrophages). We found that CD163 was elevated in CRC tumor tissues (Figure 1C, *p* = 0.0002) and positively correlated with a worse pathological stage (Figure 1F). 

### 3.2. CRC-Cell-Derived LA Promoted Tumor Malignancy with the Involvement of Macrophages

Second, to test whether LA can be induced by cancer cells under hypoxia, we cultured CRC cells (HCT116 and RKO) in a hypoxic condition for 24 h. We found that there was a significant upregulation of LA-transport-associated proteins (PGC1α, MCT1, MCT4, CD174) in CRC cells after hypoxic culture (Figure 2A,B). In addition, the concentration of LA in the CM increased in a cell-number-dependent manner (Figure 2C).

To explore the effect of LA on the biological behavior of CRC, we cultured CRC cells in a hypoxic environment for 24 h (co-cultured with macrophages or not, with 10 uM Oxamate added or not) and used a wound healing assay and a colony formation assay to reflect the changes in growth and metastasis. We found that hypoxia alone did not directly stimulate CRC cells’ proliferation and metastasis. Nevertheless, being co-cultured with macrophages in hypoxia could obviously promote CRC cells’ metastasis (Figure 2D–G) and proliferation (Figure 2H–K).) This boost was blocked by the 10 uM Oxamate (inhibitor of LA generation). These suggested that LA-mediated CRC malignancy required the engagement of macrophages.

### 3.3. LA Induced M2 Macrophages’ Polarization by Activating the AKT/ERK Pathway

In order to explore what specific effects of LA on macrophages can cooperate to promote tumor malignancy, we further treated M0 macrophages with CMs and LA at a range of concentrations, respectively. It was found that the mRNA expression of M2 macrophage markers (CD301 and TGF-β) was significantly increased in the CM- and LA-treated groups, compared with the control group (Figure 3A,B). Meanwhile, IF staining showed that the expression of CD206 (a marker of M2 macrophages) was significantly increased in the CM3-treated M0 macrophages (Figure 3C). These results suggested that LA induced the polarization of M0 macrophages to M2 macrophages.

Mechanically, we found that the phosphorylation of AKT/ERK (the downstream target of mTOR) was enhanced in a LA-concentration-dependent manner in macrophages (Figure 3D,E). MCSF, a chemokine known to stimulate macrophages’ proliferation and differentiation, was set as a positive control here (10 ng/mL). In addition, we found that a low concentration of LA (0–2 mmol/L) could not stimulate the AKT-ERK well, but high concentrations of LA (5–20 mmol/L) could activate the AKT-ERK signaling pathway significantly (Figure 3E). Oxamate, an inhibitor of LA generation, could significantly reduce the concentration of LA in the CM (10 μM) (Figure 3F), and the stimulatory effect of a high concentration of LA on the AKT-ERK signaling pathway could be counteracted by it. These findings indicated that LA in high concentrations could activate the AKT-ERK signaling pathway in macrophages, and inhibition of LA impaired the stimulatory effects of CMs on the pathway.

Next, we verified the effect of LA on tumor proliferation through macrophages in vivo. In the allogeneic subcutaneous tumor model, we found that LA injection individually was able to promote tumor growth at a moderate level (probably because there were no defects in the macrophages in the Balb/c mice); nevertheless, the simultaneous injection of LA and macrophages promoted tumor growth to a large extent (Figure 3H,I). Further, we extracted tumor tissue protein for WB and found that injecting LA into tumors mixed with macrophages could significantly activate the AKT/ERK signaling pathway in tumor tissues (Figure 3J). These confirmed the synergistic tumor-promoting effect of LA and macrophages in vivo.

### 3.4. Upon LA Stimulation, the Expression of CCL8 Was Significantly Elevated in Macrophages

TAM-derived chemokines play an important role in supporting tumor malignancy [28,29]. To explore the mechanism of how LA promoted tumor progression by regulating macrophage polarization, we next performed RNA sequencing on macrophages after treatment with 20 mM LA for 24 h. KEGG pathway analysis performed on the differentially expressed genes suggested that LA could significantly activate the cytokine- and cytokine-receptor-related signaling pathway of macrophages (Figure 4A). We further determined the chemokines that were related to the M2 macrophages, and the results revealed that the expression of CCL2, CCL7, and CCL8 increased remarkably, while CCL3 presented a decreased expression pattern (Figure 4B, Table 3). Meanwhile, we validated this phenomenon by qRT-PCR in LA-treated macrophages (Figure 4C).

Based on the result that CCL8 was the most significantly increased among all chemokines, we focused on the question of whether and how the CCL8 secreted from macrophages affects the biological behavior of CRC. We then examined the expression of CCL8 in the serum and tissues of healthy donors and CRC patients. The results suggested that the level of CCL8 in the serum of CRC patients was significantly higher than that of healthy donors (Figure 4D). In addition, IHC showed that the positive rate of CCL8+ macrophages in CRC tissues was significantly higher than that in adjacent nontumor tissues (Figure 4E,F). LA-treated macrophages released a large amount of CCL8, which itself was overexpressed in CRC patients, suggesting that CCL8 released from macrophages was closely related to CRC malignancy.

### 3.5. CCL8 Promoted Tumor Cell Proliferation and Metastasis through CCR5

CCR5, as an important receptor of CCL8, plays an essential role in promoting the proliferation and metastasis of colorectal cancer [30]. We demonstrated that the expression of CCR5 was upregulated in CRC tissues compared with adjacent nontumor tissues (Figure 5A). We then found that rCCL8 (100 ng/mL) could promote the proliferation and migration of HCT-116 and RKO cells and the expression of the skeleton proteins (F-actin and p-SMAD2)) and epithelial–mesenchymal transition (EMT)-associated proteins (PCNA, vimentin, N-cadherin), while these could be reversed by the CCR5 inhibitor Maraviroc (Figure 5B–F).

Next, in the in vitro experiments, Balb/c-nude mice (in order to exclude the interference of T cells) were used to establish subcutaneous and lung metastasis models of CT26 cells. We found that CCL8 accelerated tumor growth and lung metastasis compared to the control group, but this promotion could be blocked by the CCR5 inhibitor, Maraviroc (Figure 5G–K). These results indicated that CCL8 might accelerate the proliferation and metastasis of tumor cells through the CCL8/CCR5 axis.

### 3.6. mTOR/70S6K/4EBP1 Was a Key Downstream Pathway of CCL8/CCR5

Based on the in vitro and in vivo findings, we investigated possible downstream targets of CCL8/CCR5 activation. The Warburg effect, once thought to be due to mitochondrial damage, has been found to be regulated by mTORC1, a key regulator in mitochondrial function [31]. We found that the mTOR/70S6K/4EBP14EBP1 signaling pathway was significantly activated in a dose-dependent manner in RKO cells treated with rCCL8, while inhibiting CCR5 with Maraviroc showed the opposite effect, suggesting that the mTOR/70S6K/4EBP1 signaling pathway might be involved in the pro-cancer effects of CCL8/CCR5 (Figure 6A,B). For further verification, RKO cells were treated with CCR5 antagonists or CCR5 knocked down by shRNA. Both Maraviroc treatment and CCR5 knockdown inactivated the mTOR/70S6K/4EBP1 pathway in CCL8-treated RKO cells (Figure 6C,D). Taken together, these results suggested that the mTOR/70S6K/4EBP1 signaling pathway was a key signal axis of CCL8/CCR5 in CRC.

## 4. Discussion

Lactate, as the byproduct of glycolysis, was strictly controlled in physiological status. Recent studies showed that it not only is a cellular energy substance but also acts as a signaling molecule engaged in many diseases. In our study, we found that compared to adjacent nontumor tissues, LA-generation-related genes were significantly upregulated in CRC tissues, and they were associated with a poorer pathological stage. In addition, we discovered that LA and its transport-related genes were remarkably increased in CRC cells under hypoxia. This coincided with the previous finding that tumor cells located in hypoxic regions of solid tumors tended to generate a significant quantity of LA [32]. Surprisingly, we observed that the acidic environment under hypoxia alone does not directly affect tumor growth; the involvement of macrophages was also required.

TAMs are one of the most infiltrated cells in the TME; they are divided into the classically activated M1-like phenotype and the alternatively activated M2-like phenotype by the polarization states. M1-TAMs boost antitumor responses by presenting antigens and activating immune components [33], while M2-TAMs inhibit inflammatory responses, evade immune monitoring, and promote tumor growth [34]. The polarization of TAMs is largely determined by the stimuli of the substance from the TME. In 2014, lactic acid was confirmed to induce TAM polarization for the first time [35]. The significant role of lactic acid in tumor immune evasion has also been under investigation. Toszka found that LA-induced acidification promoted M2 polarization in melanoma [36]. In Zhang’s study, LA induced the M2 polarization of TAMs and thus promoted the invasion of pituitary adenoma [37]. In Gao’s research, LA stimulated M2 polarization and aggravated the carcinogenic behaviors of CRC cells [13]. In our study, we found that M2 macrophages were augmented in CRC tissues and positively related to a worse clinical stage. In addition, unlike those being cultured under hypoxia alone, macrophages co-cultured with CRC under hypoxia could significantly promote tumor growth and migration. Both in vivo and in vitro experiments implied that the growth and metastasis of tumors were distinctly geared up under the joint action of both LA and macrophages. Mechanically, we found that LA in high concentrations induced the M2 macrophages through the AKT-ERK signaling pathway, which is an important pathway for the maintenance of M2-type macrophages [38]. The AKT-ERK pathway in macrophages was significantly downregulated after LA inhibition.

TAM-derived chemokines played an important role in supporting tumor malignancy [28,29,39]. CCL8, also referred to as monocyte chemotactic protein 2, is associated with various immune-related pathologies [40,41]. Now, it is increasingly reported that CCL8 plays an important role in tumors. It was found that CCL8 was increased in advanced CRC patients [42] and might participate in lung metastasis [43]. In breast cancer, CCL8 was produced by TAMs, was an independent poor prognostic factor, and enhanced cancer cells’ malignancy [44]. In pancreatic ductal adenocarcinoma, CCL8 promoted the proliferation and invasiveness of cancer cells through the NF-κB signaling pathway [45]. Recently, CCL8 was identified as a hypoxic-niche factor and played a significant role in the trafficking of TAMs to hypoxic regions and reprogramming them toward immunosuppressive patterns [39]. In our study, we discovered and proved that CCL8 was one of the most increased M2-macrophage-related chemokines in macrophages exposed to LA stimulation. In CRC clinical samples, CCL8+ macrophages were enriched evidently in tumor regions. The utilization of CCL8 promoted the proliferation, EMT, and metastasis of CRC cells. CCR1, CCR2B, CCR3, and CCR5 are receptors for CCL8 that are expressed in both immune and tumor cells and play pro-cancer and anti-cancer roles [46]. CCR5 plays an essential role in promoting the proliferation and metastasis of colorectal cancer [44], but the role of the CCL8/CCR5 axis in colorectal cancer has not been clarified. In this research, we demonstrated that CCR5 was upregulated in clinical CRC tumor sites. And we substantiated that CCL8 facilitated CRC progression through CCR5; inhibiting CCR5 could block this phenomenon both in vivo and in vitro. Mechanically, we revealed that the mTOR/70S6K/4EBP1 signaling pathway was a key signal axis of CCL8/CCR5 in CRC.

## 5. Conclusions

Taken together, in this study, we found that LA produced by CRC could induce M2 polarization by activating the AKT-ERK pathway. After polarization, M2 macrophages secreted abundant CCL8 and accelerated tumor progression and metastasis through the CCR5/mTOR pathway (Figure 7). This mutual feedback and regulation between macrophages and tumor cells may be central to formulate the inhibitory niche of TME and maintain the malignancy of a tumor. Targeting the CCL8/CCR5/mTOR pathway is a promising target for future tumor treatment.

However, some limitations existed in this study. First, we did not investigate the prognostic significance of CCL8 in patients with colorectal cancer (CRC) or its relationship with clinicopathological factors. Second, besides tumor cells, TAMs also interact with other cell types in TME, including T cells, endothelial cells, fibroblasts, and so on; these form a co-evolving cancer ecosystem, ultimately determining the fate of tumors [45]. This study lays a foundation for future research on the interaction between CRC and TAM.

## Figures and Tables

**Figure 1 cancers-15-05795-f001:**
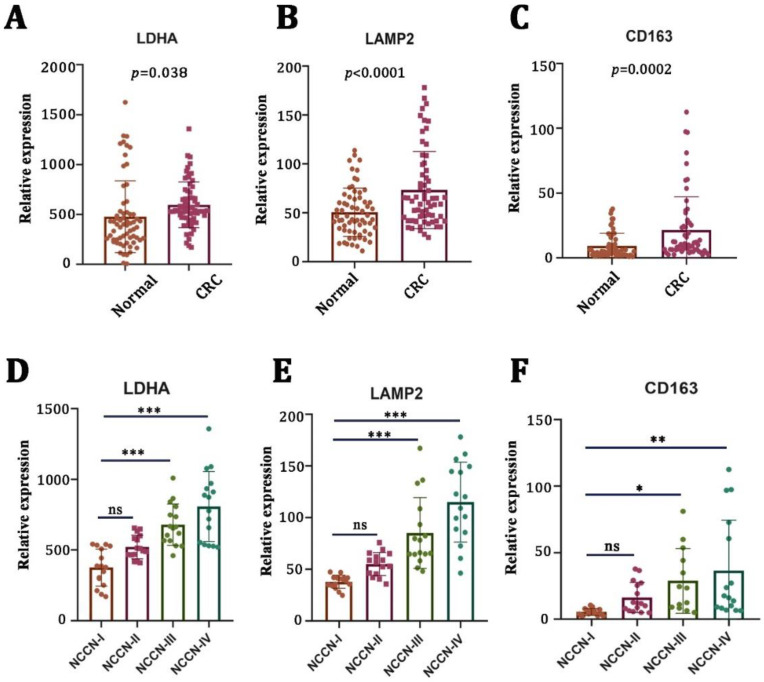
Lactate-related and M2-macrophage-related genes were upregulated in CRC and correlated with a worse pathological stage. (**A**–**C**) mRNA levels of lactate-generation-related genes (LDHA and LAMP2) and M2-macrophage-related genes (CD163) in CRC specimens (n = 64) versus nontumor tissues (n = 64). (**D**–**F**) The correlation between LDHA, LAMP2, CD163, and clinical stages of CRC patients. * *p* < 0.05; ** *p* < 0.01; *** *p* < 0.001; ns no significance.

**Figure 2 cancers-15-05795-f002:**
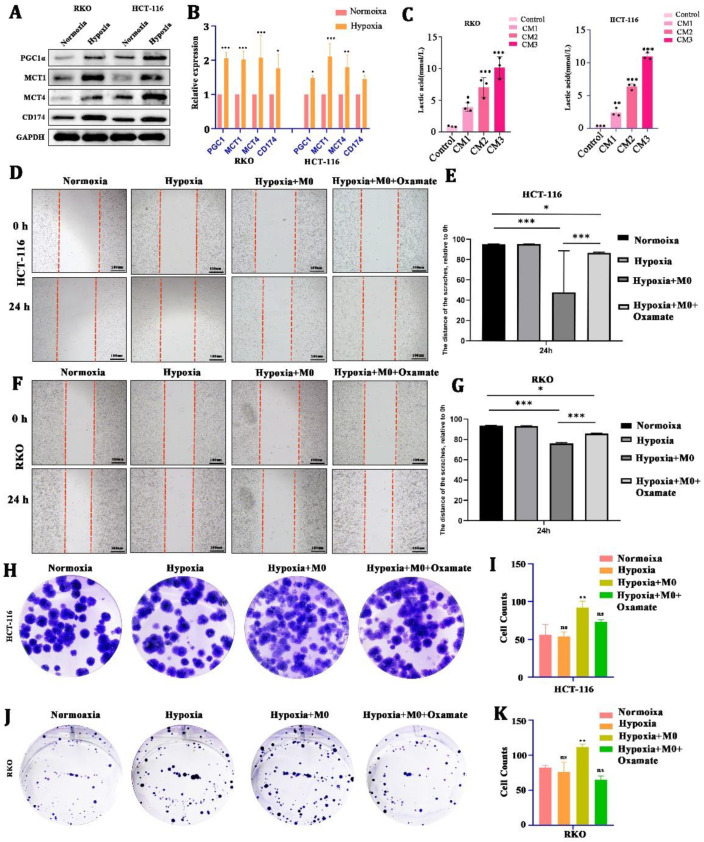
CRC-cell-derived lactate promoted tumor malignancy with the involvement of macrophages. (**A**,**B**) Protein and mRNA expression of lactate-transport-related genes of CRC cells in hypoxia or normoxia. (**C**) Concentration of lactate in culture medium from CRC cells with different cell numbers. (**D**–**G**) Representative images and statistical analysis of the wound healing assay using CRC cells cultured in normoxia, hypoxia, hypoxia +M0 macrophages, or hypoxia +M0 macrophages + Oxamate. (**H**–**K**) Representative images and statistical analysis of the colony formation assay in CRC cells cultured in normoxia, hypoxia, hypoxia +M0 macrophages, or hypoxia +M0 macrophages ++3-Oxamat. All *t*-tests were two-tailed. Mean ± SEM. CM1, culture medium from 5 × 10^5^ cells; CM2, culture medium from 1 × 10^6^ cells; CM3, culture medium from 2 × 10^6^ cells; * *p* < 0.05; ** *p* < 0.01; *** *p* < 0.001; ns, no significance. Original western blots are presented in Appendix A. Scale bar: 100 μm.

**Figure 3 cancers-15-05795-f003:**
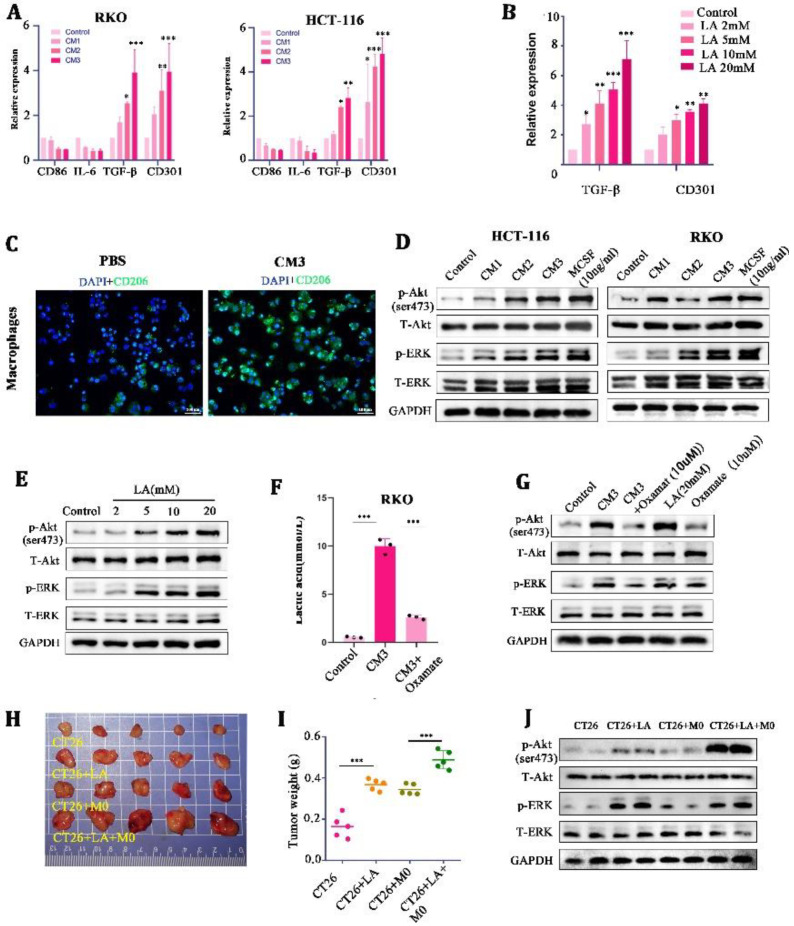
Lactate induced M2-like polarization by activating the AKT/ERK pathway in macrophages. (**A**,**B**) mRNA expression of M2 macrophage markers (CD86, IL-6, TGF-β, and CD301) in CM-stimulated THP-1 macrophages for 24 h. (**C**) Representative images of immunofluorescence in CM-treated macrophages for 24 h. (**D**) Protein expression of the AKT-ERK pathway in culture medium or M-CSF-treated THP-1 macrophages for 24 h. (**E**) Protein expression of the AKT-ERK pathway in lactate-treated THP-1 macrophages for 24 h. (**F**) Concentration of lactate in RKO cells treated with blank control, CM3, or CM3 + Oxamate. (**G**) Protein expression of the AKT-ERK pathway in THP-1 stimulated with CM3, CM3 + Oxamate, lactate, or Oxamate for 24 h. ((**H**,**I**) Representative images and tumor weight of tumor-mice inoculated with CT26, CT26+LA, CT26 + M0, and CT26 + M0 + LA. (**J**) Protein expression of the AKT-ERK pathway in tumor tissues * *p* < 0.05, ** *p* < 0.01, and *** *p* < 0.001. Original western blots are presented in Appendix A. Scale bar: 100 μm.

**Figure 4 cancers-15-05795-f004:**
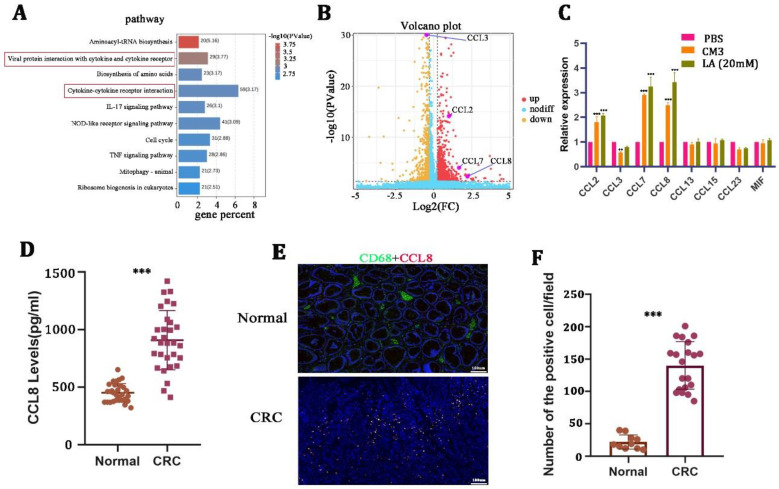
CCL8 was induced by lactate in macrophages. (**A**,**B**) KEGG pathway and volcano map and volcano map analysis of differential genes on lactate-treated macrophages for 24 h. (**C**) qRT-PCR in macrophages stimulated with CM3 or LA for 24 h. (**D**) ELISA detection of CCL8 in the serum of healthy and CRC donors. (**E**,**F**) Representative images and quantitative analysis of immunohistochemistry for CCL8 and CD68 in CRC tissues and adjacent nontumor tissues. All *t*-tests were two-tailed. Mean ± SEM. ** *p* < 0.01; *** *p* < 0.001. Scale bar: 100 μm.

**Figure 5 cancers-15-05795-f005:**
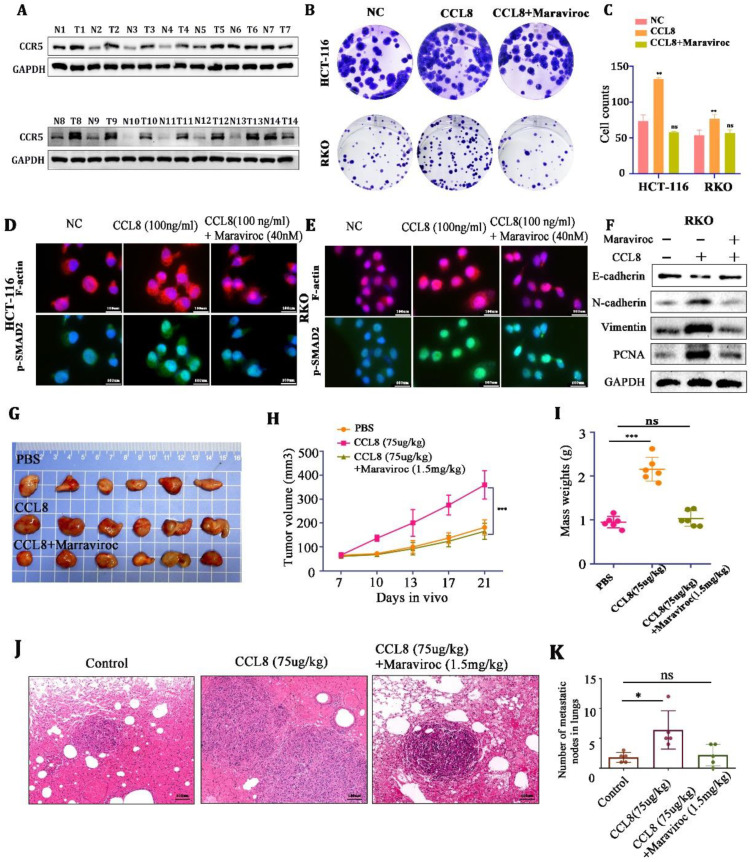
(**A**) Western blot analysis of CCR5 in CRC tissue and adjacent nontumor tissue. (**B**,**C**) Representative images and quantitative analysis of the colony assay in HCT-116 and RKO cells treated with PBS, CCL8, or CCL8+Maraviroc for 15 days. (**D**,**E**) Representative images of the immunohistochemistry for skeleton proteins (F-actin and p-SMAD2) in RKO and HCT-116 cells treated with PBS, CCL8, or CCL8+Maraviroc for 24 h. (**F**) Western blot analysis of EMT-related proteins (PCNA, vimentin, N-cadherin) in RKO cells stimulated with CCL8, Maraviroc, or both for 24 h. (**G**–**K**) Representative images and quantitative analysis of subcutaneous and lung metastasis models of Balb/c-nude tumor-bearing mice, treated with PBS, CCL8, or CCL8 + Maraviroc. All *t*-tests were two-tailed. Mean ± SEM. * *p* < 0.05, ** *p* < 0.01; *** *p* < 0.001; ns, no significance. Original western blots are presented in Appendix A. Scale bar: 100 μm.

**Figure 6 cancers-15-05795-f006:**
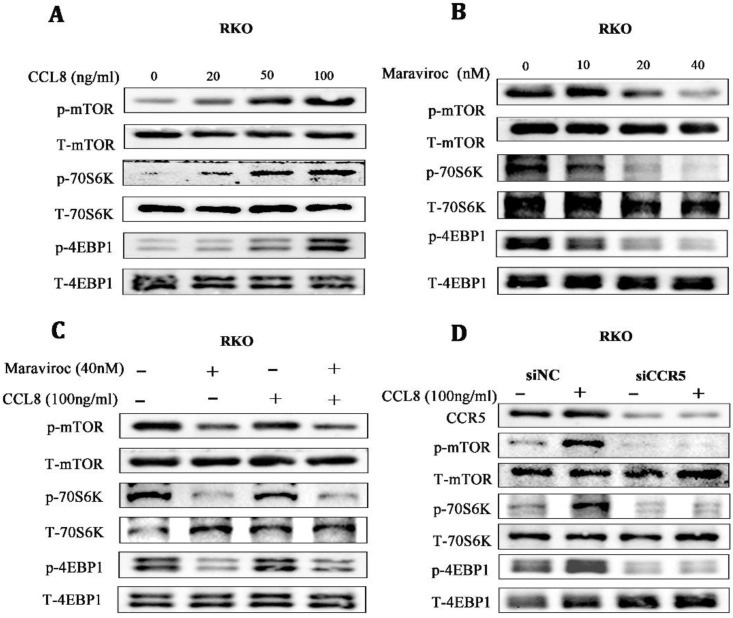
(**A**–**C**) The alterations of the mTOR/4EBP1/70S6K signaling pathway in RKO cells, after treatment with CCL8, Maraviroc, or a series of combinations of CCL8 and Maraviroc for 24 h. (**D**) The alterations in the mTOR/4EBP1/70S6K signaling pathway in RKO cells before and after CCR5 knockdown, treated with or without CCL8 for 24 h. Original western blots are presented in Appendix A.

**Figure 7 cancers-15-05795-f007:**
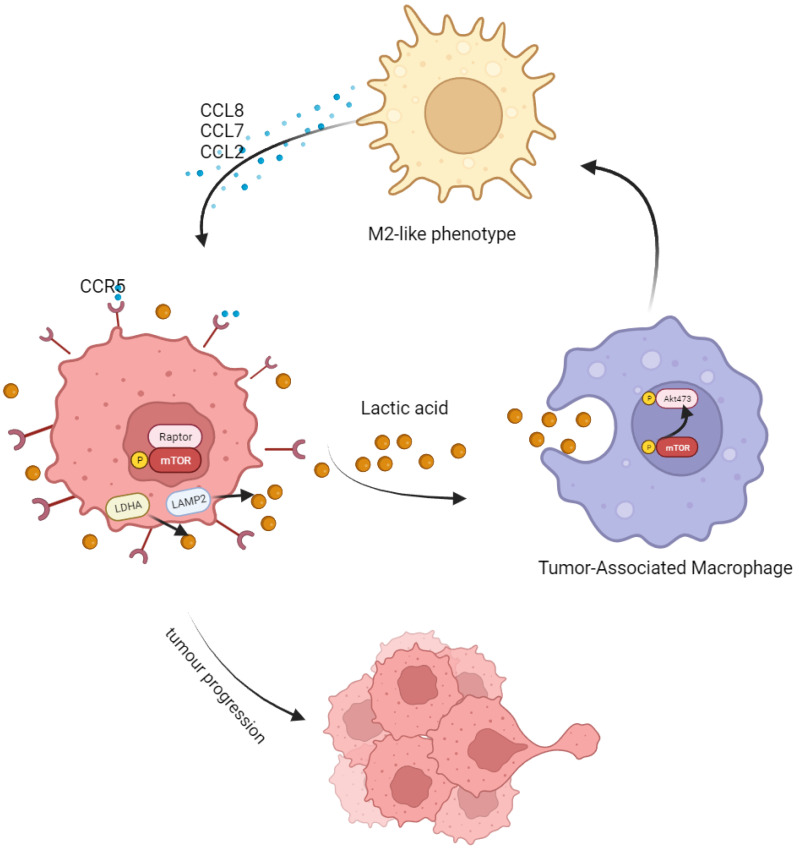
Schematic model showing the tumor-progression-promoting interaction between CRC cells and TAMs. Tumor-associated macrophages (TAMs) are more likely to adopt an M2-type polarization in response to the high lactate level in the tumor microenvironment. This leads to an increase in the secretion of CCL8 by polarized TAMs, which binds to the CCR5 receptor on tumor cells and promotes their proliferation and metastasis via the mTOR/70S6K/4EBP1 signaling pathway.

**Table 1 cancers-15-05795-t001:** Primer sequences.

Gene	Forward Sequence (5′ to 3′)	Reverse Sequence (5′ to 3′)
CD301	GTGGATGGAACAGACTATGCG	ATGGAAGTGAGCACAGTCCT
CCL2	AGAATCACCAGCAGCAAGTGTCC	TCCTGAACCCACTTCTGCTTGG
CCL3	ACTTTGAGACGAGCAGCCAGTG	TTTCTGGACCCACTCCTCACTG
CCL7	ACAGAAGGACCACCAGTAGCCA	GGTGCTTCATAAAGTCCTGGACC
CCL8	TATCCAGAGGCTGGAGAGCTAC	TGGAATCCCTGACCCATCTCTC
CCL13	GATCTCCTTGCAGAGGCTGAAG	TCTGGACCCACTTCTCCTTTGG
CCL15	TCTGGACCCACTTCTCCTTTGG	GAGTGAACACGGGATGCTTTGTG
CCL23	CCGTGTTCACTCCTGGAGAGTT	GCTTCAGCATTCTCACGCAAACC
IL-6	AGTCCTGATCCAGTTCCTGC	CAGGCTGGCATTTGTGGTTG
MIF	AGAACCGCTCCTACAGCAAGCT	GGAGTTGTTCCAGCCCACATTG
CD163	CCGGGAGATGAATTCTTGCCT	GGTATCTTAAAGGCTCACTGGG
TGFβ	GATGTCACCGGAGTTGTGCG	TGAACCCGTTGATGTCCACTTG
LDHA	GGATCTCCAACATGGCAGCCTT	AGACGGCTTTCTCCCTCTTGCT
LAMP2	GGCAATGATACTTGTCTGCTGGC	GTAGAGCAGTGTGAGAACGGCA
CD86	CCATCAGCTTGTCTGTTTCATTCC	GCTGTAATCCAAGGAATGTGGTC
Ctr siRNA	GCGGTAGGCGTGTACGGT	ATTGTGG ATGAATACTGCC
CCR5 siRNA	GUCCAAUCUAUGACAUCAATT	UUGAUGUCAUAGAUUGGACTT

**Table 2 cancers-15-05795-t002:** Antibody list.

Antibodies	Dilution	Source
GAPDH	1:1000	(Proteintech, Rosemont, PA, USA)
phosphorylated p70S6K	1:1000	(CST, Danvers, MA, USA)
p70S6K	1:1000	(CST, Danvers, MA, USA)
phosphorylated 4EBP1	1:1000	(CST, Danvers, MA, USA)
4EBP1	1:1000	(CST, Danvers, MA, USA)
phosphorylated mTOR	1:1000	(CST, Danvers, MA, USA)
mTOR	1:1000	(CST, Danvers, MA, USA)
CCR5	1:1000	(Abcam, Waltham, MA, USA)
phosphorylated Akt(S473)	1:1000	(CST, Danvers, MA, USA)
Akt	1:1000	(CST, Danvers, MA, USA)
phosphorylated ERK	1:1000	(CST, Danvers, MA, USA)
ERK	1:1000	(CST, Danvers, MA, USA)
E-cadherin	1:1000	(Santa Cruz, Dallas, TX, USA)
N-cadherin	1:1000	(Santa Cruz, Dallas, TX, USA
vimentin	1:1000	(CST, Danvers, MA, USA)
PGC1α	1:1000	(Abcam, Waltham, MA, USA)
MCT1	1:1000	(Abcam, Waltham, MA, USA)
MCT4	1:1000	(Abcam, Waltham, MA, USA)
CD174	1:1000	(Abcam, Waltham, MA, USA)
Rabbit Anti-Mouse IgG mAb (HRP Conjugate)	1:5000	(CST, Danvers, MA, USA)
Mouse Anti-Rabbit IgG mAb (HRP Conjugate)	1:5000	(CST, Danvers, MA, USA)
CD68	1:50	(Abcam, Waltham, MA, USA)
CCL8	1:100	(Abcam, Waltham, MA, USA)
CD206	1:100	Santa Cruz
Donkey Anti-Rabbit Secondary Antibody, Alexa Fluor™ 647	1:500	(Thermo Fisher. Waltham, MA, USA)
Goat Anti-Mouse Secondary Antibody, Alexa Fluor™ 488	1:500	(Thermo Fisher, Waltham, MA, USA)

**Table 3 cancers-15-05795-t003:** RNA sequencing analysis of M2-macrophage-related chemokines in macrophages treated with lactate for 24 h.

Symbol	NC-1_fpkm	NC-2_fpkm	LA-1_fpkm	LA-2_fpkm	log2(fc)	*p*-Value	FDR
CCL2	53.580	45.843	93.716	107.895	1.020	<0.001	<0.001
CCL3	1465.566	1369.561	1053.934	1018.764	−0.452	<0.001	<0.001
CCL7	1.941	2.572	7.612	6.896	1.685	<0.001	<0.001
CCL8	0.584	0.345	2.196	2.183	2.237	0.005	0.050
CCL13	<0.001	<0.001	<0.001	<0.001	<0.001	1.000	1.000
CCL15	<0.001	<0.001	0.115	<0.001	5.845	0.770	0.946
CCL23	0.100	0.300	0.215	<0.001	−0.896	0.701	0.946
MIF	125.882	129.000	134.089	126.985	0.035	0.455	0.876

## Data Availability

The data presented in this study are available on request from the corresponding author.

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
