# Peer review of "Lactate-Induced CCL8 in Tumor-Associated Macrophages Accelerates the Progression of Colorectal Cancer through the CCL8/CCR5/mTORC1 Axis"

_cancers, 2023, doi:10.3390/cancers15245795_

Round 1

Reviewer 1 Report

Comments and Suggestions for Authors

Authors have investigated the role of lactate in CRC TME, specifically on TAMs phenotype. I found the manuscript quite interesting, however, I also considered it a little bit 'messy'. Results obtained in cells, mouse models and patients are all mixed up, which complicates understanding the chronology authors followed during the experiments and the hypothesis they were considering at the different points. I strongly recommend a deep restructuration of the whole manuscript to enhance its clarity. 

Also, please revise carefully the terminology. The word lactate can be found as 'Lactate' at some points (line 70, line 88), and as 'lactate' at others. Please attach to a single nomenclature. 

Comments on the Quality of English Language

The manuscript might benefit from English Language editing, I found some paragraphs a little bit confusing. 

Reviewer 2 Report

Comments and Suggestions for Authors

The authors aimed at exploring the effect of CRC cell-derived lactate on TAMs’ polarization and the subsequent influence on CRC cells after polarization. They found that lactate produced by CRC could induce M2 polarization by activating AKT-ERK pathway. After polarization, M2 macrophages secreted abundant CCL8 and accelerated tumor progression and metastasis through CCR5/mTORc1 pathway. Albeit, I consider these findings to provide new insight into cancer-related fields, I still have some suggestions.
1, Most figures are highly professional; however, the authors should guide the readers to the meaning of the images appropriately; otherwise, it will likely cause misunderstandings. Therefore, I suggest the author consider revising these figures and legends again.

2, So far, the tumor infiltrates immune cells and is vital for patient survival. Therefore, it is worth discussing CCL8/CCR5/mTORC1 pathway correlated with immune cells by using the "TIMER" (http://timer.cistrome.org) analysis tool (PMID: 32442275).

3, In Figure 3, the author used Volcano Plot presenting that LA could significantly activate the chemokine-related signaling pathway of macrophages via RNA-seq. However, it would be much better if the author could label important genes in this plot. Meanwhile, the plot seems to have some problems with data processing; please check the data or code carefully for errors. I suggest they can try some bioinformatics tools for data visualization: http://www.bioinformatics.com.cn/srplot. Meanwhile, since Connectivity Map (CMap) can be used to discover the mechanism of action of small molecules, functionally annotate genetic variants of disease genes, and inform clinical trials. It would be fascinating if these data could be correlated with other clinical databases. Therefore, I suggest the authors can validate their data via CMap, and discuss these methodologies and literature in the manuscript  (PMID: 29195078).

4, There are few typo issues for the authors to pay attention to; please also unify the writing of scientific terms. “Italic, capital”? Please double-check superscripts and subscripts for the whole manuscript.

5, Most references are out of date, the author needs to discuss the recent paper as well as the analysis methods in this manuscript.

6, The font is too small for the current figures; meanwhile, the manuscript also needs English proofreading.

Comments on the Quality of English Language

Editing of English language required

Reviewer 3 Report

Comments and Suggestions for Authors

Manuscript Cancers-2715900

Lactate-induced CCL8 in tumor-associated macrophages, accelerates the progression of colorectal cancer through CCL8/CCR5/mTORC1 axis” for Cancers

 Comments:

1. Materials and methods. Minor corrections. Please pay attention and make corrections to any text errors, i.e. superscripts and subscripts should be corrected.

2. Materials and methods. Please complete the description with the program in which semi-quantitative densitometric analysis of the blots was performed.

3. Fig. 1 I and J. Please insert the measurement and scale bar onto images.
